# Rising slopes—Bibliometrics of mountain research 1900–2019

**Wolfgang Gurgiser**[1]*, **Martin Francis Price**[2], **Irmgard Frieda Juen**[1], **Christian Körner**[3], **Michael Bahn**[4], **Bernhard Gems**[5], **Michael Meyer**[6], **Kurt Nicolussi**[7], **Ulrike Tappeiner**[4], **Stefan Mayr**[8]

**1** Research Area Mountain Regions, University of Innsbruck, Innsbruck, Austria, **2** Centre for Mountain Studies, University of the Highlands and Islands, Perth, United Kingdom, **3** Institute of Botany, University of Basel, Basel, Switzerland, **4** Institute of Ecology, University of Innsbruck, Innsbruck, Austria, **5** Unit of Hydraulic Engineering, University of Innsbruck, Innsbruck, Austria, **6** Institute of Geology, University of Innsbruck, Innsbruck, Austria, **7** Institute of Geography, University of Innsbruck, Innsbruck, Austria, **8** Institute of Botany, University of Innsbruck, Innsbruck, Austria

* wolfgang.gurgiser@uibk.ac.at

## Abstract

Mountain areas provide essential resources for a significant proportion of the Earth's population. This study presents the development of mountain research between 1900 and 2019 based on peer-reviewed articles in English listed in Web of Science™ (WOS). We analyzed the number of publications over time, journals and scientific categories, frequent topics, and geographical distributions based on 40 mountain ranges and authors' countries as well as institutional contributions. From 1900–2019, 195k ±10% mountain research papers were published; over 50% from 2010–2019. While papers were published in more than 1000 different journals, indicating a wide range of disciplines engaged in mountain research, 94% of the papers were assigned to "Science & Technology", only <5% to "Social Sciences" and "Arts & Humanities". The most papers were written by researchers in the USA, followed by China. The number of papers per area or capita showed high variability across the investigated mountain ranges. Thus, geographically and disciplinarily more balanced research activities and better accessibility of knowledge about mountain regions are recommended.

## 1. Introduction

Mountainous areas are impressive and important landscapes of our planet. Various criteria have been used to define mountains, all adopting degrees of ruggedness to decide whether terrain is considered mountainous (Körner et al., 2011, 2017; Price et al., 2019). Depending on the inclusion of only immediate forelands (with highly urbanized areas) or even of large forelands with mega cities, global mountain terrain outside Antarctica covers between 12.5 and 27% of the Earth's land surface and has 0.5 to 1 billion inhabitants [1]. Due to their topographical complexity and climatic as well as geological variability over short distances, mountains are rich in biological and socio-cultural diversity and provide manifold ecosystem services crucial for people living both in mountains and lowlands [2–4]. For instance, around one quarter of the global population depends on fresh water, collected by mountains as atmospheric barriers and released from their huge natural reservoirs [5]. However, some of these ecosystem

**Data Availability Statement:** The bibliometric raw data used in this study are available from the Web of Science™ Core Collection (https://clarivate.com/webofsciencegroup/solutions/web-of-science-core-collection/) but restrictions apply to

the availability of these data, which were used under license for the current study, and so are not publicly available. However, the raw data can be reproduced at any licensed institution by applying the search string presented in this study in the Web of ScienceTM advanced search field. All geographical data used for this study are publicly available from the Global Mountain Biodiversity Assessment (https://ilias.unibe.ch/goto_ilias3_unibe_cat_1000512.html) or the Oak Ridge National Laboratory (https://landscan.ornl.gov/.) I can confirm that data access is available for any licensed institution.

**Funding:** The author(s) received no specific funding for this work.

**Competing interests:** The authors have declared that no competing interests exist.

services are currently at risk from global warming: also hydrological systems may be affected, with changes in the mountain cryosphere influencing the water cycle up to a global scale; and melting permafrost destabilizes mountain slopes, increasing risks of associated hazards. Furthermore, reductions in seasonal snow cover and water availability will have economic impacts on tourism, agriculture and energy production [e.g. 6–9].

The knowledge of mountain systems and current threats is based on centuries of mountain research. Early institutionalizations of mountain research at regional levels included the establishment of a joint working group on glacier research by the Swiss Alpine Club (SAC) and the Swiss Society for Natural Sciences (SNG) in the 1870s [10] and initiatives of the German-Austrian Alpine Club in the early 20th century. In 1913, the first journal (Revue de Géographie Alpine) focusing exclusively on research in the French and the entire European Alps was launched [11, 12]. At larger scales, research and institutionalization driven by attempts to increase knowledge on mountain (eco)systems were stimulated in the 1970s by the inclusion of a project on "Impact of Human Activities on Mountain Ecosystems" within UNESCO's "Man and the Biosphere" (MAB) Programme [13]. Outcomes included the establishment of the International Centre for Integrated Mountain Development and various other regional institutions and initiatives to support sustainable mountain development [14].

A specific mountain chapter in the UN Agenda 21 was another important impulse for mountain research, leading *inter alia* to the publication of a state-of knowledge book in 1997 [15] and the foundation of global mountain institutions in the 1990s and 2000s, particularly the Mountain Forum (until 2010), the Global Mountain Biodiversity Assessment Network (GMBA), the Mountain Research Initiative (MRI), and the Mountain Partnership. These institutions have been connecting mountain research, exchanging knowledge with residents, and raising awareness of mountain (eco)systems and their services from local to global levels. On a scientific level, they play an important role in e.g. coordinating or supporting comprehensive synthesis reports [16–19]. Such reports are essential in both societal and political contexts to deal with the present and future challenges in mountain regions, such as multiple stresses from climate or land-use changes [e.g. 20, 21], demographic changes [e.g.22, 23] or increasing demands for resources [e.g. 24, 25].

Ongoing, transdisciplinary and geographically comprehensive mountain research is required to successfully address future challenges at regional and global scales. A first global assessment of bibliometric data for mountain research with c. 14,000 publications [26] indicated that certain regions (and institutions) contributed disproportionately to our knowledge on mountains, while other regions were underrepresented in the literature. The applied methods did not allow detailed conclusions, such as regarding smaller mountain ranges or paper contents. It is also unknown how research activities on mountain areas developed in the last decade, although analyses of papers presented at international conferences provide snapshots [e.g., 27–29].

The current study investigates–with a more detailed thematic differentiation and an order of magnitude more data available than in Körner 2009 –the evolution of mountain research literature focusing on the period from 1990 to 2019 based on data from Web of Science TM. We extracted datasets on publications related to mountain research and analyzed detailed information associated to the individual papers. The analysis enabled quantitative insights into the development of publication activities over time, including geographical aspects, information on institutions and nationalities, scientific journals, and topics of mountain research.

## 2. Methods

The bibliometric analysis presented in this paper is based on peer-reviewed articles and review papers in the English language (hereafter summarized as "papers") listed in the "Web of

Science^TM Core Collection" 1900–2019 (hereafter called "WOS") and accessed via the "Web of Science^TM API". Using this information source, the study aims to identify the vast majority of peer-reviewed works related to mountain systems. The resulting dataset is used for a quantitative analysis of the evolution of mountain research papers, geographical and institutional characteristics of mountain research, and frequent topics, updating and extending the findings of the study by Körner et al. (2009). Such analyses allow, for example, the identification of key actors and trending topics and of less-studied mountain regions. Also, the derived dataset can be used to identify papers focusing on a specific mountain research topic and therefore could facilitate systematic reviews, despite the growing numbers of papers published per year.

## 2.1 General search query

To select literature related to mountain research in a broad context, we used the following criteria:

1. TI OR AB OR AK = "General mountain terms" OR "Range names"

2. DT = Article OR Review

3. PY = 1900–2019

4. LA = English

TI, AB and AK are paper title, abstract and authors´ keywords, DT is the type of the publication, PY is the search period, and LA the language of the paper. The "General mountain terms" in this study were "Alpine", "Mountain*" or "Montane". After checking available online resources, the "Range names" were obtained from the list of mountain ranges and subranges sorted by length in Wikipedia to also include common synonyms (such as "Rockies" for "Rocky Mountains"). Adding the range names to the general mountain terms increased the total hits by 38%. The complete search string is provided in S1 Table. For our search string, we avoided the use of the WOS Field Tag TS which additionally (to TI, AB and AK) contains the so-called "Keyword Plus ®", derived from cited literature. "Altitude" or "elevation", though often used in mountain papers, have so many other semantic meanings (from aeronautics to mechanics, e.g. elevator) that they were not considered sufficiently selective.

To make sure that the results did not exceed the allowed maximum number of results per query (100,000), the queries were separated into three temporal periods (1900–2009, 2010–2014 and 2015–2019) and the results merged in the post-processing. While the search algorithms always referred to title, keywords and abstract, only few abstracts were available for papers published before 1990. Thus, to understand respective irregularities in hit numbers, we also performed a query exclusively based on TI and AK.

Regarding the search term "Alps", excluded terms (used with NOT operator in query) were defined, which avoided hits of papers containing "Alps" as an abbreviation in medical contexts (e.g. for "Autoimmune lymphoproliferative syndrome"). These abbreviations were derived empirically by analyzing random samples of paper hits for the "Alps". Additionally, we excluded papers on "Mountain pass theorem" and "Mountain pass lemma", as these terms describe mathematical methods.

## 2.2 Mountain range query

In addition to the general survey on mountain research papers (see above), we performed individual queries for 40 ranges by applying the following criteria:

1. TI OR AB OR AK = "Range name" (OR "Synonyms" OR "Subrange name[s]")

2. DT = Article OR Review

3. PY = 1900–2019

4. LA = English

The considered range and subrange names are listed in S2 Table in the supplement. It is impossible to consider all potential ranges and subrange names, and authors use countless variants to identify ranges (local names, small-scale ranges, discipline-specific terms etc.). So we performed individual search queries for overarching range names, such as the Andes or the North American Cordillera, that included several subrange names, and for several smaller ranges from the Wikipedia list of Mountain ranges (for further details, see methods and search queries for ranges in S2 Table in the supplement).

For some ranges, we added geographically related conditions ("Southern Alps" AND "New Zealand") or exclusion terms (see S2 Table in the supplement) to avoid mishits when the same range names are used in different geographical regions (e.g. other Alps in Japan or New Zealand). In some papers, more than one range name was mentioned in the search fields. In such cases, the paper was counted for all range names included.

## 2.3 Evaluation of search queries

For all developed search algorithms, maximizing the number of identified mountain papers while minimizing false positive search hits was crucial [30]. To evaluate the accuracy of our general search query in identifying exclusively mountain papers ($\alpha$-error), we generated 3 random samples of 50 papers out of all paper hits and analyzed each paper by hand. On average, $91 \pm 2\%$ (mean ± standard error; analogue hereafter) of papers identified by the general query indeed presented mountain research.

We also checked the fraction of correct hits for the following individual ranges: for the Hindu Kush-Himalayas region, we got $96 \pm 3\%$, for the Alps $92 \pm 2\%$ and, for the Andes $97 \pm 3\%$ correct hits from 3 random samples of 50 papers. For the Pontic Mountains, with only 10 hits in total, the query resulted in 100% correct hits. All evaluations are available in S4 Table. We also tested the effect of adding additional subrange names to the queries for the Andes (e.g. Cordillera Blanca, Cordillera Real, Tamá Massif etc.) and the Alps (e.g. Mont Blanc massif, Hohe Tauern, Karwendel), as both ranges have many named subranges. We found the total hits to increase by only 1.9% and by 0.8%, which we consider as negligible.

To estimate the efficiency of our query in finding all relevant mountain papers ($\beta$-error), 7 authors of this study were asked to list 25 mountain papers of their specific field (at that time, they had no information about the study and their design). From listed papers, $79 \pm 21\%$ were included in our query results. When we consider the lowest sample value of 32% to be an outlier with a too vague definition of mountain research and the sample with the second lowest value (68% of papers included) to mismatch with our definition of mountains as rugged terrain only (see discussion), 90% of papers (of the remaining 5 samples) were included in our dataset. Submitted paper collections and respective statistics are available in S5 Table.

To summarize, our general search algorithm enabled the identification of ca. 90% of all WOS papers focusing on mountain areas, and more than 90% of the identified papers were on mountain research, which we consider as a solid basis for our literature analyses. However, it is impossible to avoid mishits, for instance when a mountain term is included in the name of a species [e.g. 31] or disease [e.g. 32] but the paper is not geographically related to mountains ($\alpha$-error). When testing the $\beta$-error, we learned that the colleagues, who defined the seven samples, had different opinions on what should be considered as mountain research. For example, several missing papers referred to the Tibetan Plateau, which was not included in our

range queries due to the absence of ruggedness (despite high elevation). Other papers were missed because they related to arctic environments. This demonstrates the difficulties of finding a common definition of mountains and mountain areas [1, 33] which is valid across scientific disciplines as well as spatial scales.

## 2.4 Data extraction from WOS

For papers identified by our queries, title, journal, authors' organization(s), country of organization(s), number of authors, year of publication, WOS category and subject(s), abstract, doctypes (Article or Review) and DOI number were extracted from the XML output obtained from the WOS API. Extraction of information on the authors´ organization (and host country) was difficult because sometimes different names were used for one organization or organization names were changed (e.g. due to merging of organizations) after publication of a paper. Furthermore, WOS lists authors´ organizations and countries separately when authors are from different departments (within one organization), but not when they are from the same department. Thus, we counted organizations and countries only once per paper, regardless of the number of authors per organization and/or country. Where available, we restrictively used the organization names from the so-called WOS "Organizations—Enhanced list" output, which assigns different name variants of organizations to one name. For authors with more than one affiliation, we counted all organizations (unless summarized under the same organization name in the "Organizations—Enhanced list") and different countries.

The assignment of several organizations to umbrella-like structures (e.g. University of California, Berkeley to University of California System) in the "Organizations–Enhanced list" unavoidably exacerbates the comparability of organizations (see discussion). In some cases (e.g. for papers with DOI 10.3390/su8090961 or 10.1130/B25500.1) one organization was associated to more than one name in the "Organizations–Enhanced list". For example, "Univ Chinese Acad Sci" is associated to "Chinese Academy of Sciences" and "University of Chinese Academy of Sciences, CAS"; "Univ London" is associated to "University of London" and "Royal Holloway University London". We then used the first association listed in the XML output. In the top 200 list of organizations (in terms of publication activity), we manually identified "umbrella-like organizations", including Universities, laboratories and other research institutions from different parts of a country/region; and "individual-university-like organizations", representing one University or research institutions at a single location.

## 2.5 Word frequencies in abstracts

To analyze terms frequently used in abstracts, we first removed most words without specific semantics, such as articles, conjunctions, pronouns, prepositions, as well as substantives, verbs and adjectives with unspecific meanings (e.g. "study", "increasing" or "significant") from text files. Remaining words were counted once per abstract and the total numbers summarized. We then manually checked the 1000 most frequent words and selected 55 words (see S6 Table) which were related to scientific topics or geographical scales. For those, we performed a frequency analysis for 5-year intervals from 1990 to 2019 (a period when abstracts were available for most papers). Plural forms of words or defined synonyms (e.g. climatic for climate) were considered, but each word, including its plural forms or synonyms, was only counted once per abstract (e.g. "climate" and "climatic" in the same abstract yield one hit for climate).

As the absolute number of word counts depend on the total number of papers, we calculated an index, which accounts for the increasing numbers of mountain papers over the study

period (trend index; $TI_{Int}$):

$$TI_{Int} = \frac{\frac{WC_{Int}}{WC_{Total}}}{\frac{PH_{Int}}{PH_{Total}}}$$

(1)

$WC_{Int}$ and $WC_{total}$ are the word counts per 5-year interval and in total (1990–2019), and $PH_{Int}$ and $PH_{Total}$ the respective paper hits. Thus, $TI_{Int}$ (1) shows relative changes of word frequencies per interval and (2) removes the trend caused by the increasing paper hits in the study period. $TI_{2015-2019} > 1$ indicates that the relative frequency of a word in the period 2015–2019 was above the average for 1990–2019, indicating increased relevance of respective topic (s). We used the same approach to investigate trends for individual ranges (WC in Eq 1 is substituted by range counts RC), countries (WC in Eq 1 is substituted by country counts CC) and organizations (WC in Eq 1 is substituted by organization counts OC).

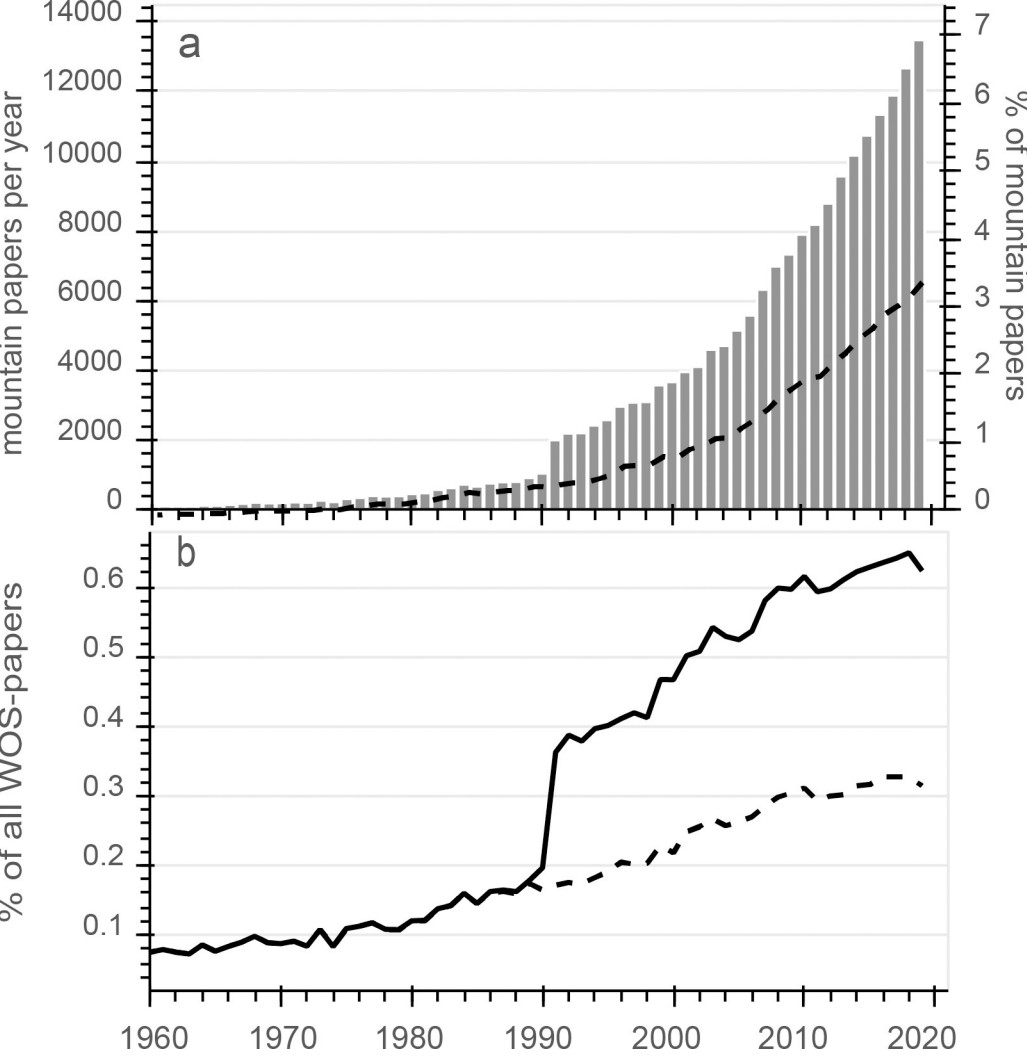

**Fig 1.** Grey bars in (a) show the number of mountain papers (left y-axis) and the % of papers per year relative to all papers (right y-axis). The dashed line indicates the number of mountain papers found in a query without searching in abstracts. In (b), % of mountain papers relative to all WOS papers of same type (English; category article or review) is shown (dashed line same as in a).

## 2.6 Geographical data

For the 40 mountain ranges considered in our queries, polygons were derived from the supplement material (GMBA data) in Körner et al. (2017). They were directly taken from GMBA data version v1.1. as single polygons (29), from version v1.2. large scale data where multiple polygons were merged to a major mountain range (6), or manually merged from single polygons in version v1.1. (5; e.g. North American Cordillera). Details are listed in S2 Table and visualized in S1 Fig. The mountainous area [defined by ruggedness of the terrain, see 34] within these polygons is derived by intersecting the mountain polygons with the pixels classified as rugged terrain using the GMBA ruggedness mountain data and definition [based on 35; Release 4 based on the digital elevation model GMTED2010, version May 2021].

Population data for 2019 were derived from LandScan 2019™ High Resolution Global Population Data Set with a resolution of 30 arc seconds (the same as the GMBA ruggedness data). By intersecting the population data with GMBA data, the population densities for the mountainous (rugged terrain only) area and the polygon area were calculated (differences are shown in S2 Fig).

## 3. Results

### 3.1 Number of mountain papers

The general search query yielded 195,114 mountain papers in the period 1900–2019. Based on our evaluation, this number should include less than 10% of false positive hits and miss less than 10% of papers relevant for mountain research as defined in this study. In 1991, a sudden increase in hits was observed, which was caused by the inclusion of abstracts in WOS datasets from this year on. When abstracts were excluded from the query over the entire period, the number of papers increased continuously (see dashed lines in Fig 1).

Within the study period, a strong increase in the number of mountain papers per year was observed (Fig 1A). In the last 10 years of the study period (2010–2019), more than half (54%) of all identified papers were published. Besides the strong increase in absolute numbers, the number of mountain papers relative to total WOS papers (Fig 1B) increased by a factor of 3 from the 1960s (<0.1%) to the 2010s (ca. 0.3%; for search without looking into abstracts). For the first half of the twentieth century, few papers are listed in WOS and therefore, the proportion of mountain papers in relation to total papers strongly fluctuated between 0.05% and 0.3%. For the period 1991–2000, when abstracts in WOS were available and more papers could be identified, the average proportion of mountain papers was 0.4% of all WOS papers; for 2010–2019, the proportion increased to 0.6%.

### 3.2 Journals, paper categories and subjects

Overall, we found 2,100 journals which published at least 10 mountain papers between 1900 and 2019 (Fig 2). The top journal (Tectonophysics) contained 2253 papers (1% of all identified mountain papers); 16 journals published more than 1000 papers; 399 journals more than 100; and 707 journals more than 50 papers. 97% of all papers were categorized as articles, 3% as reviews.

For some journals, almost all papers were published between 2015 and 2019 (Fig 2, orange bars). These journals (e.g. Scientific Reports, Remote Sensing or Forests) reached high numbers of mountain papers in a comparably short time. Also, two journals exclusively addressing mountain research were found in the top 100: Mountain Research and Development (MRD) and the Journal of Mountain Sciences (JMS) were at ranks 14 and 29 for the period 1900–2019, and at ranks 42 (MRD) and 9 (JMS) for the period 2015–2019, demonstrating a strong

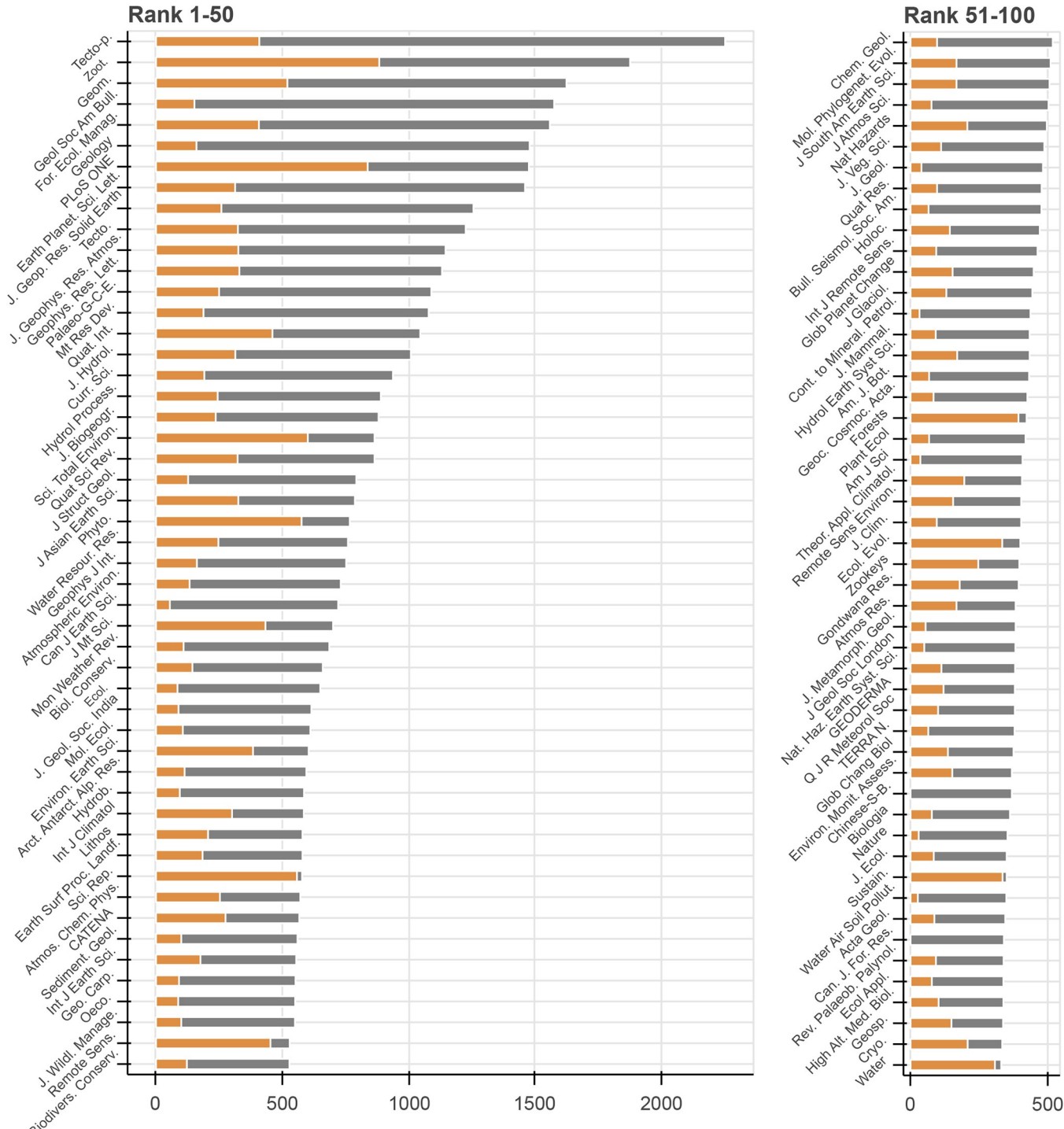

**Fig 2. Number of mountain papers published in journals for 1900–2014 (grey end) and for 2015–2019 (orange end).** The full names associated with the used abbreviations are available in the supplement (S3 Table).

increase in recently published papers in JMS. The Revue de Géographie Alpine–Journal of Alpine Research, as mentioned in the introduction, was at place 149 whereas only the papers in English language were counted.

**Table 1. The 10 most frequent WOS traditional subjects of mountain papers with absolute hits and relative share (%).**

|  | Counts | % |
|---|---|---|
| Geosciences: Multidisciplinary | 36443 | 12 |
| Environmental Sciences | 21676 | 7 |
| Ecology | 21376 | 7 |
| Geochemistry & Geophysics | 15461 | 5 |
| Plant Sciences | 14646 | 5 |
| Geography: Physical | 14295 | 5 |
| Meteorology & Atmospheric Sciences | 13352 | 4 |
| Zoology | 10610 | 4 |
| Water Resources | 9228 | 3 |
| Geology | 8529 | 3 |

More than 94% of the papers were assigned to the field of "Science & Technology", only 4 and 2% to "Social Sciences" and "Arts & Humanities". The most frequent paper subject (Table 1) was "Geosciences: Multidisciplinary" (12%), followed by "Environmental sciences" and "Ecology" (7% each); several papers were assigned to more than one subject category. These statistics of the papers' subjects highlight that the great majority of papers were related to natural sciences.

### 3.3 Terms in abstracts

The most frequent term in abstracts was "change(s)", which was found in 21% of all analyzed papers between 1990 and 2019 (Fig 3), often combined with the words "climate" or "climatic" (32% of all cases), "environmental" (10%), "global" (8%), "economic" (3%), and "social" or "societal" (2%). Notably, the word combination "climate change" was only included in 5% of

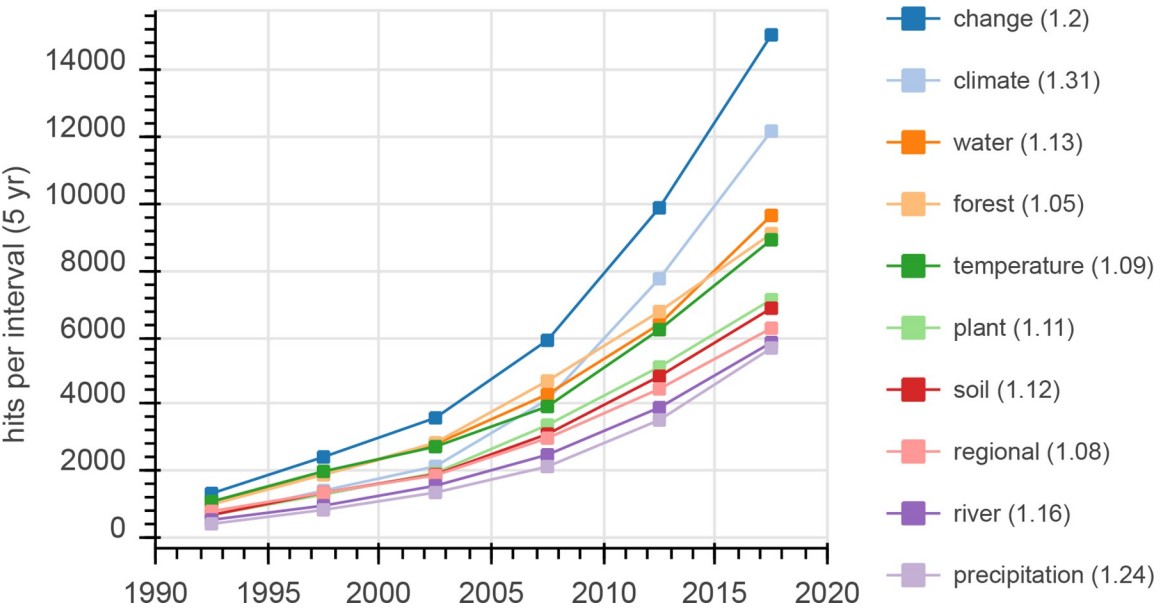

**Fig 3. Number of abstract hits that contain a certain word for 5-year intervals from 1990–2019.** Number in brackets is the Trend Index (see Eq 1) for 2015–2019.

all abstracts, reflecting the different options to describe the same or similar phenomena (e.g. "climatic change(s)", "changes in climate" etc.).

16% of all analyzed abstracts contained the word "climate" or "climatic", and the number of papers on climate-related issues increased over time relative to other topics (Fig 3). Next in frequency were the terms "forest(s)" (15% of all abstracts) and "water" (14% of all abstracts). The overall number of papers related to water in a broader sense was higher as, for example, only 31% of abstracts including "river(s)" also included "water".

The term "temperature(s)" was found in about 14% of the abstracts. Respective papers were often related to climate topics (42% of the abstracts with "temperature(s)" also included climate-related terms). "Plant(s)" and "soil(s)" were included in around 11% and 10% of all abstracts, and 25% of the abstracts with "plant(s)" also contained "soil(s)". "River(s)" and "tree (s)" were both included in more than 8% of the abstracts; "tree(s)" and "forest(s)" co-occurred in 59% of all cases. Notably, frequencies of the 10 most frequent words between 1990 and 2019 increased above the average between 2015 and 2019.

The term "regional(ly)" is different from the other words in the top 10 list as it describes a geographical context (not a topic). The term appeared in 10% of all abstracts and more often than "global(ly)" (7% of all abstracts). However, "global(ly)" had a $TI_{2015-2019}$ of 1.35, indicating that studies including global contexts have been increasing in absolute and relative frequencies in recent years. However, any interpretation has to consider non-geographical uses of "globally".

Other trending terms were "hydropower" ($TI_{2015-2019}$ = 1.74), "biodiversity" ($TI_{2015-2019}$ = 1.49) and "ecosystems" ($TI_{2015-2019}$ = 1.41): their absolute frequency in papers from 2015 to 2019 differed by 1 or 2 orders of magnitude (for details, see S7 Table—WordsTop10).

## 3.4 Mountain ranges

Most papers (see Table 2 and Fig 7) were assigned to the Hindu Kush-Himalayas (HKH) region (21,412), the European Alps (16,314), the Andes (12,992), and the North American Cordillera (10,879). Regarding papers per 1,000 capita per mountainous area (referencing to rugged terrain only) or per polygon area (number in brackets; see Fig 4A), the Alps had a value of 2.1 (0.9), the North American Cordillera of 1.3 (0.1), the Andes of 0.5 (0.1) and the HKH region of 0.4 (0.2). Relative to the mountainous or polygon (number in brackets) area (Fig 4B), the Alps had the highest ratio of papers per $km^2$ of all ranges 0.1 (0.09). For the HKH region, the ratio was 0.01 (0.006), for the Andes 0.007 (0.004), and for the North American Cordillera 0.005 (0.003). The proportion of paper contributions from authors based in local countries to total authors (Table 2) varied considerably, with 37% for the Andes, 56% for the HKH region, 73% for the Alps, and 83% for the North American Cordillera.

In addition to the major ranges mentioned above, the Carpathians, Apennines, Appalachian Mountains, Western Ghats, and Qin Mountains had >1,000 absolute paper hits. Ratios of paper numbers vs. 1,000 capita were between 7.1 and 0.1 (Fig 4A). The differences between paper numbers per capita per mountainous vs. polygon area were pronounced for some of these ranges (e.g. the Appalachian Mountains in Fig 4A). Regarding the ratio of paper numbers vs. area, values were in the order of 0.01 (Fig 4B). The percentage of authors based in local countries (Table 2) was rather high (73–92%) for all these ranges.

Regarding the number of papers per mountain range since 1990 (Fig 5), we observed the strongest increase for the HKH region, which was above the general trend of published mountain papers in the period 2015–2019 ($TI_{2015-2019}$ = 1.16). For the same period, paper contributions increased slightly above average for the Andes ($TI_{2015-2019}$ = 1.04), the Carpathian Mountains ($TI_{2015-2019}$ = 1.16), the Western Ghats ($TI_{2015-2019}$ = 1.32), and Qin Mountains

**Table 2. Total hits per mountain range and percentage of paper authors based in local countries (countries that at least have some overlap with the range area).**

|  | Range | Considered Subranges | Total Hits | % local Authors |
|---|---|---|---|---|
| 1 | Hindu Kush-Himalayas region | Himalayas, Sivalik Hills, Hindu Kush, Karakoram, Hengduan Mountains, Arakan Mountains | 21412 | 56 |
| 2 | Alps | Western Alps, French Prealps, Eastern Alps, Central Eastern Alps, Northern Limestone Alps, Southern Limestone Alps, Western Limestone Alps | 16314 | 73 |
| 3 | Andes | Cordillera Occidental, Cordillera Central, Cordillera Oriental, Precordillera | 12992 | 37 |
| 4 | North American Cordillera | Rocky Mountains, Coast Mountains, Peninsular mountain ranges, Brooks Range, Sierra Madre del Sur, Alaska Range, Sierra Nevada, California Coast Ranges, Cordillera Occidental, Cordillera Oriental | 10879 | 83 |
| 5 | Carpathian Mountains |  | 4771 | 74 |
| 6 | Apennines |  | 3947 | 74 |
| 7 | Appalachian Mountains |  | 3413 | 91 |
| 8 | Western Ghats |  | 2617 | 76 |
| 9 | Qin Mountains |  | 1971 | 78 |
| 10 | Dinaric Mountains |  | 738 | 53 |
| 11 | Caucasus Mountains | Greater Caucasus, Lesser Caucasus | 675 | 47 |
| 12 | Cascade Range |  | 608 | 92 |
| 13 | Southern Alps |  | 603 | 47 |
| 14 | Zagros Mountains |  | 476 | 0 |
| 15 | Drakensberg |  | 476 | 65 |
| 16 | Atlas Mountains |  | 476 | 0 |
| 17 | Altai Mountains |  | 438 | 52 |
| 18 | Ural Mountains |  | 403 | 25 |
| 19 | Serra do Mar |  | 293 | 79 |
| 20 | Scandinavian Mountains |  | 247 | 52 |
| 21 | Kunlun Mountains |  | 239 | 76 |
| 22 | Kopet Mountains |  | 220 | 52 |
| 23 | Taurus Mountains |  | 195 | 48 |
| 24 | Great Dividing Range |  | 113 | 86 |
| 25 | Balkan Mountains |  | 82 | 43 |
| 26 | Japanese Alps |  | 76 | 83 |
| 27 | Barisan Mountains |  | 41 | 37 |
| 28 | Aravalli Range |  | 33 | 80 |
| 29 | Espinhaco Mountains |  | 19 | 73 |
| 30 | Eastern Sayan Mountains |  | 18 | 75 |
| 31 | Annamite Range |  | 11 | 33 |
| 32 | Pontic Mountains |  | 10 | 17 |
| 33 | Koryak Mountains |  | 10 | 0 |
| 34 | Mantiqueira Mountains |  | 8 | 94 |
| 35 | Verkhoyansk Range |  | 6 | 38 |
| 36 | Chersky Range |  | 5 | 50 |
| 37 | Suntar-Khayata Range |  | 4 | 17 |
| 38 | Kolyma Mountains |  | 2 | 25 |
| 39 | Dzhugdzhur Mountains |  | 0 |  |
| 40 | Stanovoy Highlands |  | 0 |  |

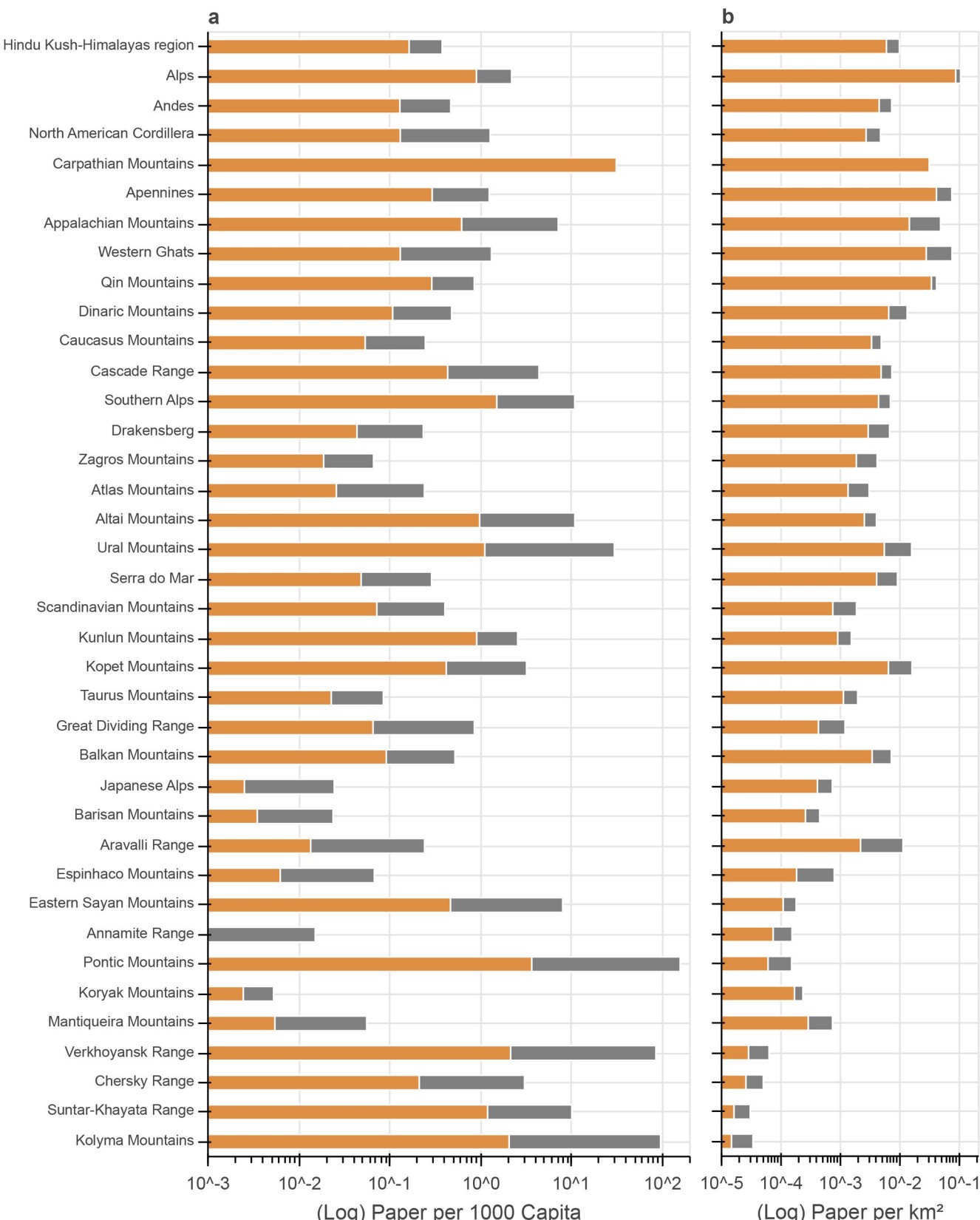

**Fig 4. Paper hits per capita (a) and per area (b; logarithmic scales) for all ranges.** Ranges are sorted based on the number of total paper hits. Ranges with no hits are not shown. Orange refers to the polygon area, orange plus grey to the mountainous area (see methods). Capita data represent 2019 values.

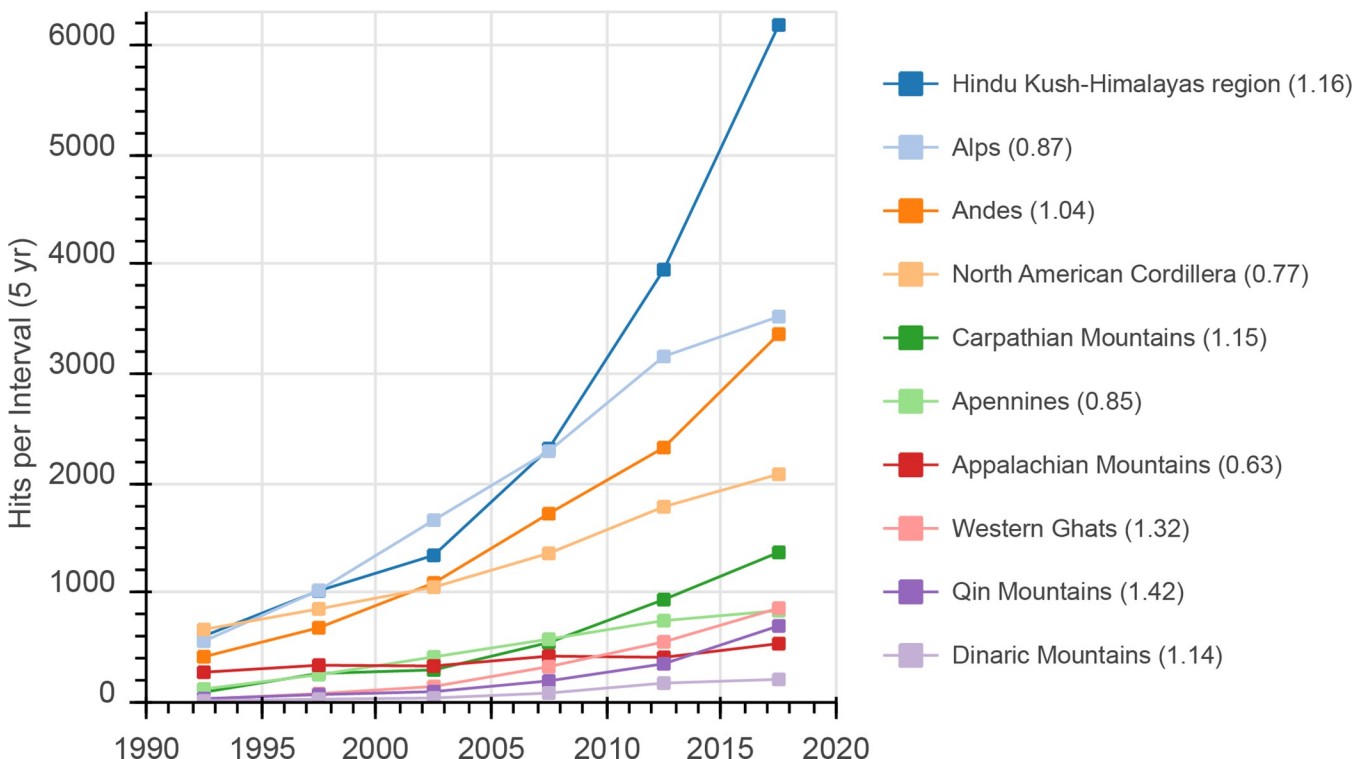

**Fig 5. Number of hits for the Top 10 ranges for 5-year intervals from 1990–2019.** Number in brackets is the Trend Index (see Eq 1) for 2015–2019.

($TI_{2015\text{-}2019}$ = 1.43). In contrast, for the Alps and the North American Cordillera, despite increasing absolute numbers of paper contributions from 2015–2019, the increase was below the general trend ($TI_{2015\text{-}2019}$<1). For the Apennines and the Dinaric mountains, the absolute paper contributions increased slightly in this period, while $TI_{2015\text{-}2019}$ was still >1 for the latter. Some mountain ranges with low numbers of total contributions but strong increases in recent years were the Altai ($TI_{2015\text{-}2019}$ = 1.51), Kopet Mountains ($TI_{2015\text{-}2019}$ = 1.48) and the Qin Mountains ($TI_{2015\text{-}2019}$ = 1.42; for more details, see S7 Table—MountainRangesTop10).

### 3.5 Countries

For the entire period (1900–2019), most paper contributions came from researchers based in the USA, China, Germany, Italy and UK (Figs 6A and 7). However, Fig 6A shows that the ranking has changed in recent years and India reached rank 3 in the period 2015–2019. Fig 6A also highlights a strong increase in paper contributions from China, which reached rank 2 in 2010–2015 with a clear above average relative increase in the period 2015–2019 ($TI_{2015\text{-}2019}$ = 1.48). In contrast, some other countries, such as the USA, Canada and Switzerland, showed a slight decrease in relative contributions.

Table 3 shows country-specific publication activities (period 2015–2019) including values corrected for population and economic data. Normalized by population, authors affiliated at institutions in Switzerland published the most papers, followed by other countries with a small area but large proportion of mountainous area (e.g. Austria, Iceland, Andorra) or with low population density (e.g. Norway or Iceland). Normalized by GDP ($) per capita as economic index, India reached the highest values, followed by China and Nepal. Countries such as Myanmar, Lithuania or Kazakhstan had very low numbers of total contributions to mountain

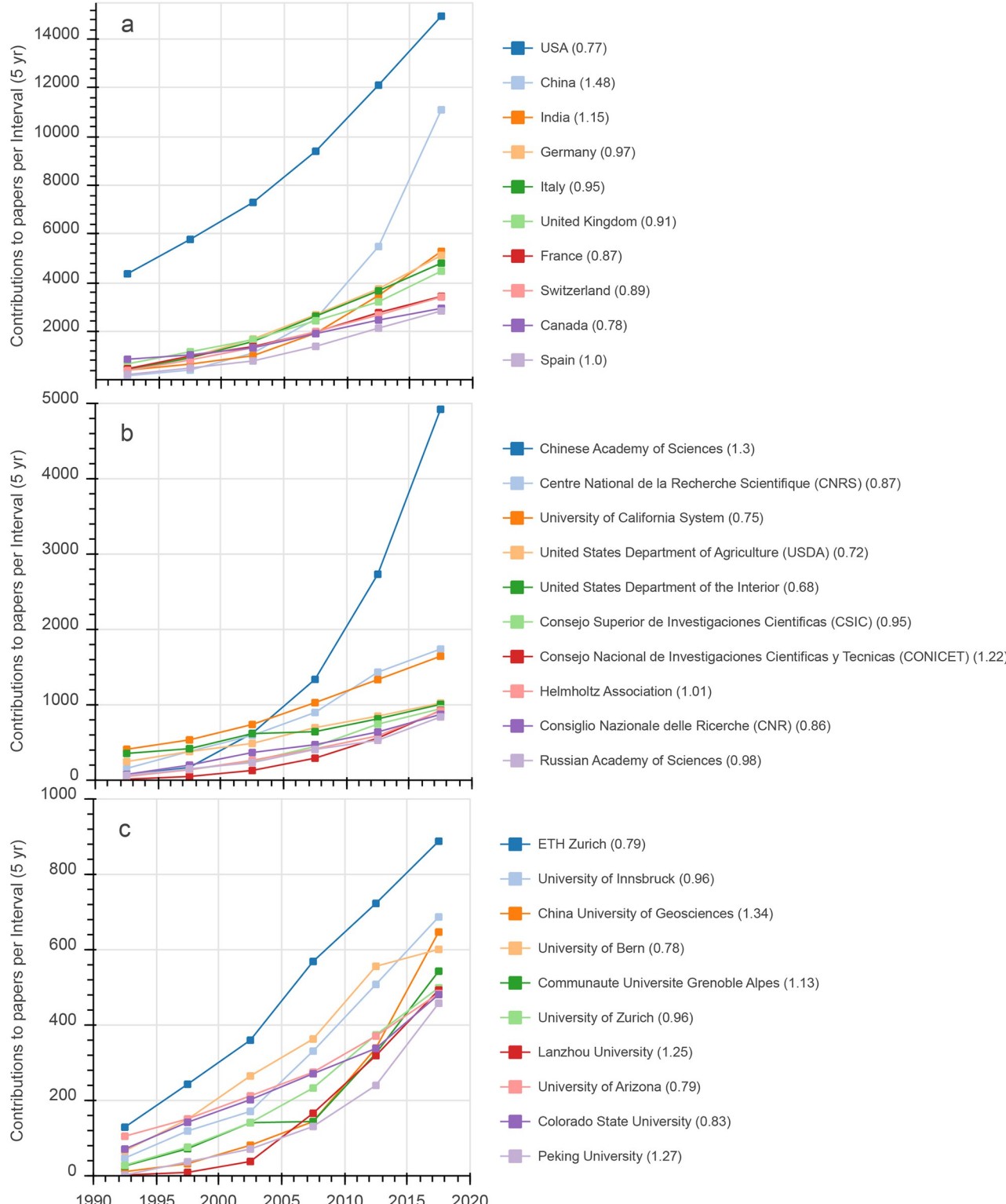

**Fig 6.** (a) Top 10 Country contributions to papers for 5-year intervals from 1990–2019. Paper contributions of (b) Top 10 umbrella-like and (c) Top 10 individual-like organizations for 5-year intervals from 1990–2019. Number in brackets are the Trend Index values (see Eq 1) for 2015–2019.

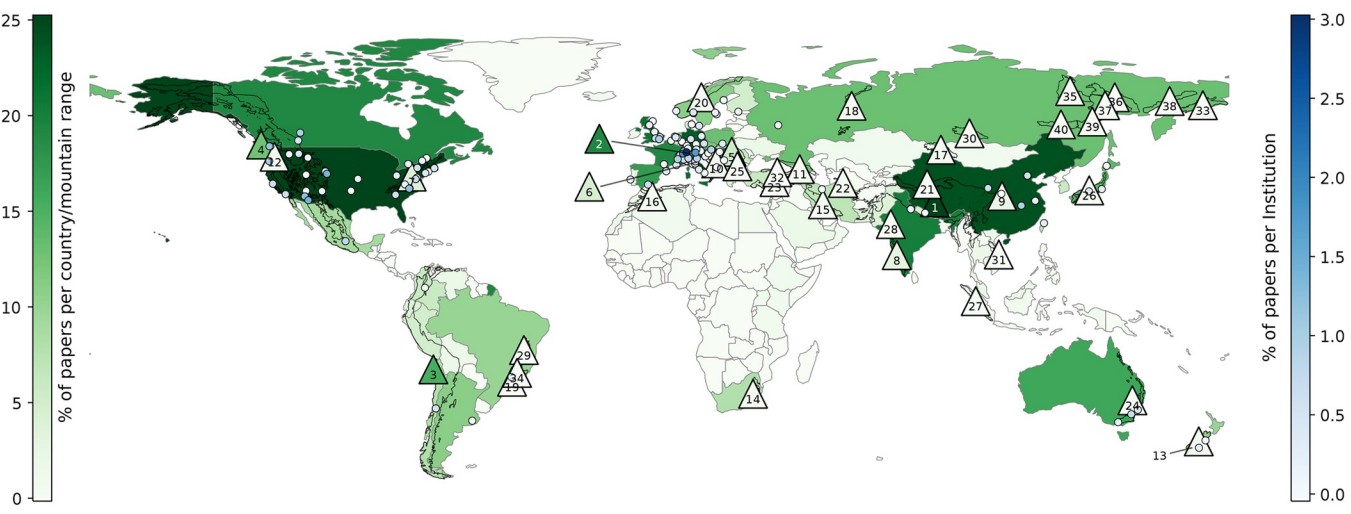

**Fig 7. % of total papers per country (greens) and range (greens; triangles with numbers referring to mountain ranges in Table 2).** % of total papers per university-like institution (blues; dots) for the top 200 university-like institutions. Polygons (black edge-color) indicate the considered mountain ranges (see methods and discussion for details). Background image made with Natural Earth.

research but notable increases in recent years (for more details, see S7 Table—CountriesTop10).

## 3.6 Organizations

With respect to umbrella-like organizations, the Chinese Academy of Sciences reached the top position in 2005–2010 and has since become, by far, the largest contributor to mountain research papers, with more than twice as many contributions as the CNRS (France) or the University of California System (Fig 6B). These three organizations also ranked 1 to 3 for the entire period (1900–2019). With the USDA and the USDI at positions 4 and 5, there were three organizations from the USA in the top 10, although these three organizations showed clearly decreasing trends in relative contributions in the period 2015–2019 ($TI_{2015\text{-}2019} < 1$). The three academic consortia from Europe in the top 10 were the CSIC in Spain, the Helmholtz Association in Germany, and the CNR in Italy. These organizations showed a slight decrease or a stable proportion of contributions ($TI_{2015\text{-}2019}$ ca. 1). The other organizations in

**Table 3. Top 10 results of country paper contributions absolute, per Mio. capita (average values for period) and per GDP ($) per Capita (GDPpC; average values for period) for the period 2015–2019.** Exact values in brackets.

| Papers | Papers per Mio. Inh. | Papers per GDPpC |
|---|---|---|
| USA (14950) | Switzerland (403) | India (2.80) |
| China (11108) | Austria (234) | China (1.23) |
| India (5280) | Iceland (212) | Nepal (0.78) |
| Germany (5119) | Norway (196) | Pakistan (0.48) |
| Italy (4804) | Andorra (194) | USA (0.25) |
| United Kingdom (4471) | Slovenia (177) | Iran (0.20) |
| France (3435) | New Zealand (155) | Ethiopia (0.18) |
| Switzerland (3409) | Liechtenstein (132) | Russia (0.16) |
| Canada (2939) | Czech Republic (128) | Brazil (0.16) |
| Spain (2834) | Slovakia (121) | Italy (0.15) |

the top 10 were CONICET (Argentina: position 7, increasing trend) and the Russian Academy of Sciences (position 10, stable trend).

With respect to individual, university-like organizations, the ETH Zurich was on rank 1 for both 1900–2019 and 1990–2019, although with a decreasing trend ($T_{2015\text{-}2019}$ = 0.79) in relative paper contributions in recent years (Fig 6C). From 2015–2019, the University of Innsbruck was ranked second, with an almost stable trend in relative contributions. At rank 3 was the China University of Geosciences, with the strongest increase in absolute and relative paper contributions. It was followed by the University of Bern, which showed a marked decrease in relative contributions in the last interval. In contrast, the Université Grenoble Alpes showed slightly increasing relative contributions from 2015–2019 and reached rank 5. The Universities of Zurich and Arizona, Lanzhou and Peking Universities, and Colorado State University had similar total contributions in the last interval but different trends.

S7 Table—UmbrellaLikeOrganizationsTop10 and UniLikeOrganizationsTop10—presents some newly emerging institutions in mountain research.

## 4. Discussion

The large dataset available in the Web of Science<sup>TM</sup> core collection enabled reliable insights into the evolution of mountain research activities from 1900 to 2019, particularly since the 1990s. Based on the presented search query, nearly 200,000 mountain-related English peer-reviewed articles could be extracted and analyzed regarding development of publication activities, journals and scientific categories, frequent topics, geographical distributions with respect to mountain ranges, authors' countries and institutions.

The share of mountain papers from total papers in WOS was below <0.1% for decades but almost doubled between 1980 and 1990 (Fig 1). One of the main reasons may be UNESCO's MAB Programme, which greatly stimulated mountain research [36]. From 1991 to 2019, the share further increased by around 70%. On one hand, this reflects the success of activities (such as the establishment of global mountain research institutions; see introduction) following the MAB Programme. On the other hand, research on climate change in various contexts disproportionally increased in these years [e.g. 37, 38] and such research has become the most frequent topic in recent mountain research (Fig 3), underlining the global importance of mountain areas with respect to all aspects of climate change (Adler et al. 2022).

Overall, the increasing quantity of papers indicates that the status of mountain research and knowledge of various mountain-related topics have substantially improved in recent years [as e.g. investigated in detail by 38]. According to the analysis of frequent terms (Fig 3), many present-day topics and challenges, as mentioned in the introduction of this study or in the recently published cross chapter on mountains in IPCC AR6 (Adler et al., 2022), have been intensively addressed, at least from a natural-sciences perspective (disciplinary imbalances are further discussed below). Our analysis also shows that the number of contributions dealing with emerging challenges, e.g. the increasing demand for renewable energy resources from mountain systems, particularly from hydropower, has grown markedly in recent years (S7 Table–WordsTop10).

However, the increase in papers makes it more challenging to keep an overview of the mountain research output, discuss potentially diverging results, and identify general and more specific research gaps. Noteworthy in this context, the proportion of reviews within mountain research papers decreased from 4.1% for 2000–2009 to 2.4% for 2010–2019. Datasets derived when applying the search queries presented in this study may be a helpful source to facilitate the preparation of future in-depth analyses and synthesis products dealing with present and future issues in mountain regions.

The large number of journals publishing mountain research reflects the broad range of scientific disciplines involved (Fig 2), although most of the identified top 100 journals focus on natural sciences. In recent years, a shift in the number of papers from journals with long traditions in mountain research towards newly emerging journals took place. Thorough investigations of the trends in disciplinary contributions, based on the datasets presented in this study, would be desirable to e.g. find ways to deal with increasing numbers of interdisciplinary works and journals. Nevertheless, the findings indicate that the imbalances between natural and social sciences, evident in the past and recently [13, 15, 19, 26, 39] remain. These imbalances impose the risk that information from, and approaches in, natural sciences are not well linked to societal needs. Therefore, there is a continuing need for support for interdisciplinary projects and studies that are conducted by, or involve, social scientists (Gleeson et al. 2016, Price et al. 2022).

Crucially, the WOS restriction to papers in the English language, as already noticed by Körner (2009), means that a substantial fraction of mountain papers (e.g. in Chinese, French, German, Japanese, Russian or Spanish) are less recognized and are also missing from this study. From the perspective of the international mountain research community, it is highly desirable that relevant findings are (also) published in English-language journals and that institutions engaged in fostering mountain research are encouraged to provide support to researchers who are not native English speakers, such as editorial support (a long-standing policy of MRD) or translation services.

Analyzing geographical aspects as summarized in Fig 7, the four major ranges, the Hindu-Kush Himalayas region, the Alps, the Andes and the North American Cordillera, received 73% of total hits from the range queries, with increasing numbers of absolute contributions over time (Fig 5 and Table 2), which demonstrates the importance of these geographic areas within global mountain research. Overall, there was a high variability in total paper hits, as well as hits related to capita and area. Several ranges, such as the Japanese Alps and ranges in the east of Russia, had remarkably low ratios relative to capita and/or area, indicating limited information about these ranges, at least in English, WOS-listed journals. One outcome of this study could be to stimulate local and other researchers to check the current state of knowledge regarding these mountain ranges and highlight research deficits in a geographical context or make local (traditional) knowledge available for the international research community. In addition, the absolute quantity of papers and associated ratios per area/capita only reflects one of numerous aspects regarding the state of knowledge of individual mountain ranges. For example, exchanges between local societies and researchers are crucial for knowledge generation and application [40]. In this context, the Andes, the Ural Mountains, and the Atlas Mountains, for example, had exceptionally low ratios of paper authors based in local countries (Table 2) which we consider disadvantageous for discourse among actors. These aspects should be considered in future national and international research strategies to reduce imbalances.

Considering contributions of countries, the USA and China have by far the highest absolute numbers of published mountain papers, with China showing an impressive increase during the last two decades (Fig 6A and Table 3). China´s paper output related to its economic capacity (estimated by GDP per capita; Table 3) was also high, demonstrating its outstanding engagement in mountain research. It should be noted that other Asian countries, such as India, Nepal or Pakistan, also showed remarkable engagement in mountain research relative to their economic capacity. Per capita, Switzerland remains the leading contributor [26], followed by other European countries and New Zealand, demonstrating the long tradition and profound experiences in mountain research in these mostly mountainous countries. South America, despite its large mountain areas, showed relatively low research activity: Brazil was the only country in the top ten of any of our statistics presented in Table 3 (which fits also to

the low share of local paper authors observed for the Andes; see above). Given the high importance of mountains for many countries in South America, this might be a motivation to check whether substantial amounts of scientific works are only available in Portuguese or Spanish [41] and, as recognized by the Conéctate+ initiative of MRI, to increase international cooperation to *inter alia* educate or train local researchers, and to strengthen mountain research institutions. Similarly, in Africa only one country (Ethiopia due to low economic capacity) was among any of the top ten groups and, based on the results for Atlas Mountains (Table 2) and [42], we assume that the involvement of local researchers has been below average. Similar recommendations could therefore be made for Africa, where the Southern African Mountain Conference has recently provided an opportunity to bring together researchers. Our results for South America and Africa are in line with earlier findings [28].

With respect to institutions (Fig 6), results regarding umbrella-like organizations have to be interpreted with care: the analysis is clearly biased by the varying sizes of these organizations (a political/economic rather than a scientific aspect), with a striking example being the overwhelming size of Chinese Academy of Science network. In the USA and the European Union, such umbrella-like organizations are smaller, though several were ranked within the top ten. For individual, university-like organizations, strongly increasing contributions from Chinese universities caused a slight decreasing trend in relative shares for all other universities, except the Université Grenoble Alpes. In absolute numbers, ETH Zurich maintained its dominant position in mountain research over the years [26], followed by the University of Innsbruck, which recently reached rank two.

## 5. Conclusions

The overall increase in mountain research activities and new topical hot spots in mountain research are promising developments with respect to new challenges such as global changes, as already called for in the first global strategy for research in mountain areas, published in 2005 [43]. The increase in papers underlines the value of review papers and synthesis reports (e.g. Adler et al., 2022; Egan and Price, 2017; Haddaway et al., 2020; Hock et al., 2019; Romeo et al., 2020; Wester et al., 2019) as well as the importance of mountain research institutions in strengthening capacity, especially to address disciplinary and geographic imbalances; transferring knowledge to political processes; and supporting transdisciplinary research and exchange with local societies. Accordingly, good visibility and easy accessibility of papers on mountain research are highly desirable, which should be supported by including general mountain terms and (overarching) mountain range names in the title, abstract or keywords of any paper. For all institutions actively engaged in developing mountain research further, developing strategies to monitor and reduce geographical, social and disciplinary imbalances is highly desirable.

## Supporting information

**S1 Table. General search string as applied in the Web of Science$^{TM}$ advanced search.**
(XLSX)

**S2 Table. Search strings for individual mountain ranges as applied in the Web of Science$^{TM}$ advanced search.**
(XLSX)

**S3 Table. List of journals that published identified mountain research papers with number of papers found for the period 1900–2019 and 2015–2019.**
(XLSX)

**S4 Table. List of papers defined as mountain research by several authors of this study and evaluation results.**
(XLSX)

**S5 Table. Subsamples of results from queries with quality checks.**
(XLSX)

**S6 Table. List of selected words (including synonyms) analyzed within this study.**
(XLSX)

**S7 Table. Results of Trend Index as defined in this study for words, mountain ranges, countries and organizations.**
(XLSX)

**S1 Fig. Interactive world map with polygons of the 40 mountain ranges considered in this study.** Range names are shown when moving the mouse over the polygons.
(HTML)

**S2 Fig. Differences due to geographical reference.** Difference in population density of mountain ranges calculated for the entire polygon area and only for the rugged terrain within the polygon area.
(TIF)

## Acknowledgments

We thank the editor and Graham McDowell as well as one anonymous reviewer for their constructive comments that helped to substantially improve the manuscript.

## Author Contributions

**Conceptualization:** Wolfgang Gurgiser, Stefan Mayr.

**Data curation:** Wolfgang Gurgiser, Martin Francis Price, Irmgard Frieda Juen, Christian Körner, Michael Bahn, Bernhard Gems, Michael Meyer, Kurt Nicolussi, Ulrike Tappeiner.

**Formal analysis:** Wolfgang Gurgiser, Irmgard Frieda Juen, Stefan Mayr.

**Investigation:** Wolfgang Gurgiser, Stefan Mayr.

**Methodology:** Wolfgang Gurgiser, Stefan Mayr.

**Visualization:** Wolfgang Gurgiser.

**Writing – original draft:** Wolfgang Gurgiser, Stefan Mayr.

**Writing – review & editing:** Martin Francis Price, Christian Körner, Michael Bahn, Bernhard Gems, Michael Meyer, Kurt Nicolussi, Ulrike Tappeiner.

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
