## [Decision Letter · Decision Letter 0]

16 May 2022

PONE-D-22-09392Rising slopes - bibliometrics of mountain research 1900-2019PLOS ONE

Dear Wolfgang and collaegues,

Thank you for submitting your manuscript to PLOS ONE. After careful consideration, we feel that it has merit but does not fully meet PLOS ONE’s publication criteria as it currently stands. Therefore, we invite you to submit a revised version of the manuscript that addresses the points raised during the review process.

We have now received two reviewer reports. You will see that both reports are positive and see a need for minor revisions before publication. Their comments can be addressed in a straightforward way. Among the other comments, please consider the points raised by reviewer 1 about the inclusion of an introductory paragraph in the methods section, and adding a map depicting mountain areas globally.

In addition to the reviewer reports I have a few points from an editor’s perspective. The analysis of the vast amount of mountain literature over a long time period is impressive and allows you to produce multiple quantitative analyses. I think the analytical part and interpretation of the results in the bigger mountain research context could still be strengthened in the manuscript. I like how you address several aspects in this context in the Discussion and Conclusion section. However, I think there is more room for analytical text that goes beyond reporting the metrics and numbers of the quantitative analysis. For instance, what does this impressive literature analysis actually tell un in terms of status and development of mountain research? How has mountain research developed over this long time period, based on your results and additional consideration of literature? Is current mountain research fit to address the major challenges of today and the future? I’m aware that there is some limitation of going into much detail in this regard but I think some additional analysis would really strengthen the paper, also in view of the long-term experience and expertise of the author team. In this regard, please also consider and integrate the recently published mountain chapter in the IPCC AR6.

Technically, I think that figure visibility and readability should be improved, e.g. for Figs. 2 and 7 (in the latter one the blue color range is not visible), and possibly also for Figs. 3, 5, 6.

We look forward to receiving your revised manuscript.

Kind regards,

Christian Huggel

Academic Editor

PLOS ONE

Journal Requirements:

3. We note that Figure 7 and Striking Image in your submission contain [map/satellite] images which may be copyrighted. All PLOS content is published under the Creative Commons Attribution License (CC BY 4.0), which means that the manuscript, images, and Supporting Information files will be freely available online, and any third party is permitted to access, download, copy, distribute, and use these materials in any way, even commercially, with proper attribution. For these reasons, we cannot publish previously copyrighted maps or satellite images created using proprietary data, such as Google software (Google Maps, Street View, and Earth). For more information, see our copyright guidelines: http://journals.plos.org/plosone/s/licenses-and-copyright.

a) You may seek permission from the original copyright holder of Figure 7 and Striking Image to publish the content specifically under the CC BY 4.0 license.  

Reviewers' comments:

Reviewer's Responses to Questions

**Comments to the Author**

1. Is the manuscript technically sound, and do the data support the conclusions?

Reviewer #1: Yes

Reviewer #2: Yes

2. Has the statistical analysis been performed appropriately and rigorously? 

Reviewer #1: Yes

Reviewer #2: Yes

3. Have the authors made all data underlying the findings in their manuscript fully available?

Reviewer #1: Yes

Reviewer #2: Yes

4. Is the manuscript presented in an intelligible fashion and written in standard English?

Reviewer #1: Yes

Reviewer #2: Yes

5. Review Comments to the Author

Reviewer #1: Dear Authors,

Thank you for the opportunity to review the manuscript titled ‘Rising slopes - bibliometrics of mountain research 1900-2019’.

This well written paper, which was prepared by a group of eminent mountain researchers, provides important insights into the state of mountain research globally. It begins with an accessible overview describing the importance of mountain (eco)systems as well as significant milestones in the evolution of mountain research. This is an appropriate start, given the broad readership of PLOS One. Then, based on a well implemented bibliographic review and associated quantitative analyses, it sheds new light on the nature of mountain research according to a variety of relevant metrics (e.g. publishing activity over time, the distribution of research according to mountain geographies, research activity by institutions and nationalities, leading scientific journals, and primary research approaches and topics). I was particularly struck by the authors’ tactful and well justified approach to determining the cogency of their sample considering nearly 200,000 returns, which precluded a more typical inclusion/exclusion review process. Relatedly, the paper is accompanied by helpful and interesting supplementary materials. The article, which builds upon earlier review efforts by Körner (2009), concludes by describing study limitations (e.g. focus on English language documents) as well as a number of salient research/publishing needs and opportunities.

This impressive review paper makes fundamental contributions to our understanding of the state of mountain research globally. Given the far-reaching socio-ecological importance of mountain (eco)systems, the authors’ findings should be of interests to scholars working across a range of geographies and thematic areas.

I have one substantive recommendation for improvement, as well as a few minor points for the authors’ consideration:

1. The main blind-spot in the manuscript is the lack of an introductory paragraph in the methods section that provides background on the chosen review method. For example, why was a bibliometric analysis selected over a systematic scoping review, how does such an approach differ, and why was it the most appropriate option given your study objectives? Adding such an introductory paragraph will round out an otherwise excellent methods section. This is my substantive recommendation.

2. The methods section does not mention a specific focus on ‘peer-reviewed’ articles/reviews. While this is indicated elsewhere in the manuscript, it seems prudent to state in the methods section.

3. A map depicting mountain areas globally based on the criteria described in lines 222 – 230 would provide helpful geospatial context, particularly for readers not familiar with the global distribution of mountains. Depicting the 40 mountain ranges examined in your analysis would likely be too visually complex but could be considered. Perhaps such a map could be added at line 235.

4. The phrase “economic power”, which appears throughout the manuscript, is slightly off putting. Could another word replace ‘power”, perhaps “capacity”?

5. The observed paucity of research in the social sciences seems highly consequential, yet no specific call for more engagement with the social sciences in mountain research is provided in the Discussion and Conclusion section (lines 416-417). Perhaps it would be germane to include a line about the potential implications of this research gap, as well as the need to support such research in mountainous contexts?

6. Line 420-421 rightly suggest that it “is highly desirable that relevant findings are (also) published in English-language journals.” Should this recommendation also be accompanied by a call for the mountain research community/mountain organizations to provide support for such publishing (e.g. the establishment of a fund to support translation services)?

Thank you again for the opportunity to review the manuscript titled ‘Rising slopes - bibliometrics of mountain research 1900-2019’. I believe that, with minor revisions, the study will be an important contribution to scholarly work published in PLOS One.

Best regards,

Graham McDowell

Reviewer #2: Excellent and highly relevant manuscript with great insights. Suggest to address several minor comments, which have been uploaded as a separate attachment and are also listed below:

p.1, line 24: replace “would be desirable” with “is recommended”.

p.2, lines 63-64: for comprehensive synthesis reports please consider also citing the HKH Assessment report (Wester et al., 2019) and the Cross-Chapter Paper Mountains in IPCC AR6 WGII (Adler et al., 2022).

p.9, lines 274- 276: that only 4 and 2% of the papers were assigned to the fields “Social Sciences” and “Arts & Humanities” is a significant finding. Considering highlighting this in the Abstract.

p.17, line 451: use of the word “local” languages seems inappropriate here. I imagine a lot has been published in Spanish, which is a global language. Would be good to clarify this.

p.17, line 470: For synthesis reports, consider also citing the HKH Assessment report (Wester et al., 2019) and the Cross-Chapter Paper Mountains in IPCC AR6 WGII (Adler et al., 2022).

Note: references need to be carefully checked, e.g., the first reference by Adhikari, Shrestha and Shakya is incomplete.

6. PLOS authors have the option to publish the peer review history of their article (what does this mean?). If published, this will include your full peer review and any attached files.

Reviewer #1: **Yes: **Dr. Graham McDowell

Reviewer #2: No

---

## [Author Response · Author response to Decision Letter 0]

12 Jul 2022

Please see ResponsestoReviewers_final.pdf for our comments on the review.

---

## [Editor Report · Decision Letter 1]

9 Aug 2022

Rising slopes - bibliometrics of mountain research 1900-2019

PONE-D-22-09392R1

Dear Wolfgang Gurgiser,

We’re pleased to inform you that your manuscript has been judged scientifically suitable for publication and will be formally accepted for publication once it meets all outstanding technical requirements.

Kind regards,

Christian Huggel

Academic Editor

PLOS ONE

Additional Editor Comments (optional):

please check the reference list (some references such as Adler et al. 2022 are incorrect, cf. official IPCC citations for this case).

---

## [Editor Report · Acceptance letter]

14 Aug 2022

PONE-D-22-09392R1 

Rising slopes - bibliometrics of mountain research 1900-2019

Dear Dr. Gurgiser:

I'm pleased to inform you that your manuscript has been deemed suitable for publication in PLOS ONE. Congratulations! Your manuscript is now with our production department. 

Kind regards, 

on behalf of

Dr. Christian Huggel 

Academic Editor

PLOS ONE